Out of control mortality matters: the effect of perceived uncontrollable mortality risk on a health-related decision

Pepper Gillian V. g.pepper@ncl.ac.uk
Nettle Daniel
Newcastle University, Institute of Neuroscience , Newcastle Upon Tyne , UK
Andersson Gerhard
Electronic publication date: 2014 Jun 26
Publication date: 2014
Volume: 2
Electronic Location ID: e459
Received 2014 Apr 3; Accepted 2014 Jun 8
Copyright: © 2014 Pepper and Nettle
Copyright year: 2014
Copyright holder: Pepper and Nettle
License: This is an open access article distributed under the terms of the Creative Commons Attribution License, which permits unrestricted use, distribution, reproduction and adaptation in any medium and for any purpose provided that it is properly attributed. For attribution, the original author(s), title, publication source (PeerJ) and either DOI or URL of the article must be cited.
License URL: https://creativecommons.org/licenses/by/4.0/

Keywords: Control, Mortality risk, Perceptions, Health, Behaviour, Decisions

Funding: Newcastle University Institute of Neuroscience PhD studentship This work was funded as part of a Newcastle University Institute of Neuroscience PhD studentship. The funders had no role in study design, data collection and analysis, decision to publish, or preparation of the manuscript.

==============================
Prior evidence from the public health literature suggests that both control beliefs and perceived threats to life are important for health behaviour. Our previously presented theoretical model generated the more specific hypothesis that uncontrollable, but not controllable, personal mortality risk should alter the payoff from investment in health protection behaviours. We carried out three experiments to test whether altering the perceived controllability of mortality risk would affect a health-related decision. Experiment 1 demonstrated that a mortality prime could be used to alter a health-related decision: the choice between a healthier food reward (fruit) and an unhealthy alternative (chocolate). Experiment 2 demonstrated that it is the controllability of the mortality risk being primed that generates the effect, rather than mortality risk per se. Experiment 3 showed that the effect could be seen in a surreptitious experiment that was not explicitly health related. Our results suggest that perceptions about the controllability of mortality risk may be an important factor in people’s health-related decisions. Thus, techniques for adjusting perceptions about mortality risk could be important tools for use in health interventions. More importantly, tackling those sources of mortality that people perceive to be uncontrollable could have a dual purpose: making neighbourhoods and workplaces safer would have the primary benefit of reducing uncontrollable mortality risk, which could lead to a secondary benefit from improved health behaviours.

Introduction

It is important to understand what factors influence health behaviour. Some of the leading causes of death in developed countries result from preventable unhealthy behaviours such as inactivity, poor diet, smoking and alcohol consumption (Mokdad et al., 2004). Such preventable behaviours also cause a substantial burden on healthcare systems. For example, obesity-related health problems, such as type 2 diabetes and heart disease, are becoming a major issue in the UK, with 61% of adults and 30% of children in England being overweight or obese. Such obesity and overweight related health problems are estimated to cost the NHS over £5 billion a year (Report, 2011).

A substantial research effort has been made towards improving the efficacy of health messages to promote behaviour change. One of the key ideas to emerge from this research has been that perceived control and efficacy should influence health behaviour. Health Locus of Control describes the extent to which a person believes that their health is determined by the actions of individuals, rather than by chance, and whether the locus of that control is internal (a result of their own actions) or external (resulting from the actions of others). Prior findings suggest that Health Locus of Control is important both for health outcomes (e.g., Burker et al., 2005; Poortinga, Dunstan & Fone, 2008) and for health behaviours (Reitzel et al., 2013; Wardle & Steptoe, 2003).

Other research themes focus on the effects of mortality salience and perceived threat on health behaviour. Terror Management Theory (Greenberg, Pyszczynski & Solomon, 1986) proposes that people have a fear of death which causes anxiety or terror when they are made aware of their vulnerability. It suggests that, when people are made to think about their mortality (a condition known as mortality salience) they will attempt to buffer their anxieties and to suppress conscious thoughts of death. Goldenberg & Arndt (2008) extended Terror Management Theory to create the Terror Management Health Model for behavioural health promotion. They proposed that conscious thoughts about death (as elicited by many fear appeals) would trigger behavioural responses (in this case, health improving behaviour) aimed at reducing the threat, and thus the accompanying fear of death. They proposed that when thoughts about death are unconscious, people should act not to reduce the threat to their life, but to direct their efforts to maintaining a sense of meaning and self-esteem.

The fear appeal literature combines elements of control with those of threat. (Fear appeals are messages intended to persuade people to change their behaviour by inducing fear regarding health threats.) Theoretical frameworks used in the fear appeal literature (e.g., Extended Parallel Process Models and Protection Motivation Theory-comprehensively reviewed by Witte & Allen, 2000) emphasise the importance of efficacy in eliciting behaviour change. In general, these theories suggest that if there is a strong threat to health and a highly effective solution is available, then people will act to use that solution. However, if messages offer threats without suggesting that there are effective solutions, behaviour change will not occur. That is, these models state that threat serves to motivate people towards possible solutions, but that if people do not feel that the solutions will be effective, they are unlikely to act (Goei et al., 2010; Lewis, Watson & White, 2013; Witte & Allen, 2000).

The Uncontrollable Mortality Risk Hypothesis

Similarly, our previously presented theoretical model (Nettle, 2010) combined elements of control and threat to life. It suggested that differences in health behaviour could be explained by differential exposure to uncontrollable mortality risk: the Uncontrollable Mortality Risk Hypothesis. The hypothesis suggests that people who are likely to be killed by factors beyond their control should be less motivated to invest effort in looking after their future health. This makes intuitive sense when you consider that people who are exposed to high uncontrollable mortality risk are less likely to survive to reap the rewards of their healthy behaviour, which are likely to be garnered in the far future. To give a caricatured example, there is little point in investing in a healthier diet when you feel you could be killed by an erupting volcano at any moment. We previously tested predictions from this hypothesis using survey data (Pepper & Nettle, in press). We found that people who perceived a higher portion of their personal mortality risk to be beyond their control were less motivated to invest effort in looking after their health.

Our hypothesis differs from theories in the fear appeal literature, since these focus on the controllability of the specific aspects of health which are being communicated and not on the controllability of mortality risk more generally. For example, they predict that the belief that you can control your risk of diabetes by modifying your diet will affect your motivation to eat healthily. By comparison, our hypothesis predicts that perceived control over mortality risk should alter motivation towards healthy behaviour—even when the healthy behaviour is not a recommended response to that risk. For example, if you believe you are unable to control your risk of falling victim to a volcanic eruption, you should be less inclined to eat healthily. A healthy diet is not a recommended response to reduce the threat posed by a volcano and yet, we should expect the controllability of one risk to influence the payoff to investing in mitigating the other.

Our hypothesis also takes a different perspective to Health Locus of Control studies, which tend to implicitly assume that Health Locus of Control is a stable individual trait, rather than a flexible response to information from the environment. By comparison, behaviour as a response to environmental cues is a key assumption of the Uncontrollable Mortality Risk Hypothesis. Finally, while Terror Management Theory emphasises the importance of mortality per se, our hypothesis suggests that it is the controllability or the mortality risk that should be important.

In summary, a range of theories emphasize the importance of mortality salience and control in the behavioural responses to health messages. Our Uncontrollable Mortality Risk Hypothesis specifically predicts that cueing mortality risk per se will not affect health behaviours, but rather, that it will be the controllability of the mortality risk that influences the decision to behave healthily.

Here, we present three experiments testing this prediction. The first was a test of whether mortality primes can be used to influence a health-related decision—the choice between a healthy food reward and an unhealthy one. The second experiment used the same method but with primes that separated out the effects of controllability from those of mortality priming. That is, we tested whether there is an effect of mortality salience per se, or whether it is the controllability of mortality risk which is important. The third study aimed to rule out the possibility that the results of the first two studies were due to demand characteristics; the participants did not know that they were taking part in an experiment and health was never explicitly mentioned.

Experiment 1: The Effect of Uncontrollable Mortality on a Health-Related Decision

Experiment 1 tested whether an uncontrollable mortality prime would affect a simple health-related decision: the choice between a reward of fruit (the healthy option) and chocolate (the unhealthy option). For this proof-of-concept experiment, we chose primes that we expected to produce the most extreme results. One prime suggested that causes of death were uncontrollable, and that people sharing the participant’s demographics were dying younger than average (uncontrollable short life prime). The other prime suggested both that causes of death were controllable and that people sharing the participant’s demographics were living longer than average (controllable long life prime). We predicted that participants would report stronger intentions towards healthy behaviour and be more likely to choose fruit in the controllable long life treatment than in the uncontrollable short life treatment.

Methods, materials and analysis

All of our experiments (1, 2 & 3) received ethical approval from the Newcastle University Faculty of Medical Sciences ethics committee. Participants for experiments 1 and 2 were recruited using the Crowdflower crowdsourcing platform (http://crowdflower.com). Participants followed a link to the experiment, which was generated using Qualtrics (version 2013, http://www.qualtrics.com). Participants were presented with an information screen which contained statements about ethics and privacy and provided contact details for the experimenters. The introduction to the study explained that it was about life expectancy differences within the UK (see questionnaire in Supplemental Information). This included a link to a news article about Public Health England’s Longer Lives website (http://longerlives.phe.org.uk/), which provides a map of the regions of England, ranked by rates of premature mortality. Since experiment 1 was launched on 2nd July 2013, less than a month after this map had been headline news, it made a timely cover story for the experiment. Participants completed an electronic consent form.

We needed to ensure that our participants were from the UK, because the primes were based on UK postcode statistics. Thus, participants were filtered through a location check using their Internet Protocol address (IP address) and an explicit question about whether they were resident in the UK. Participant location information (based on IP address) and reported postcode were triangulated with self-reported UK residency to assess the reliability of the data. Consistency of location reporting was used as an inclusion criterion (see Supplemental Information for full details).

Participants moved on to a screen which asked for their age, gender and current postcode. After giving this information, all participants were presented with a “loading” animation, timed to auto-progress after 12 s. The message under the animation read, “Thanks for submitting your information. It may take a while to match it to health data for people of your age and gender in your postcode area. Please wait a few moments”. This loading screen was designed to create the impression that the demographic information given by participants was being used to look up real information about life expectancies for people who shared their characteristics. Participants then were randomly allocated to one of the primes.

In each prime, the message fed back to the participant used dynamically generated content to display a message tailored with the age, gender and postcode which had been entered previously. This was done to make the participants feel as though the information about their mortality risk was personal to them.

Uncontrollable short life prime

The uncontrollable short life priming screen read as follows: “Statistics indicate that, on average, [age] year-old [male/female]s in your postcode area [(postcode)] die 13 years younger than [male/female]s of the same age in the rest of the UK. The reasons for this are unclear and may be due to factors beyond individual control, such as traffic accidents and air pollution. We want to understand more about why this is happening. Please answer the following questions about your health”.

Controllable long life prime

The controllable long life priming screen read: “Statistics indicate that, on average, [age] year-old [male/female]s in your postcode area [(postcode)] live 13 years longer than [male/female]s of the same age in the rest of the UK. The reasons for this are unclear and may be due to individual behaviours, such as diet and exercise habits. We want to understand more about why this is happening. Please answer the following questions about your health”.

Outcome variables

Following the priming screen, participants moved on to the health behaviour questions. They were asked to answer some simple scale-based (0–100) questions about their intended health behaviour over the coming week (see Supplemental Information for full questionnaire). We refer to the answers to these as self-reported health intentions. The first was a general question about the effort the participant intended to put into looking after their health. The second question was about whether the participant intended to eat the recommended 5 portions of fruit and vegetables a day. The third question was about whether the participant would do a recommended level of exercise. The final question was about how much alcohol the participant intended to consume. After the questionnaire was completed, participants were moved onto a screen, which was ostensibly separate to the questionnaire. They were thanked for taking part in the study and told that, as an extra thank you for taking part, they could opt to be entered into a prize draw. They were asked to select the prize which they would prefer to win. The options were an organic fruit box worth £11, or chocolate collection box worth £11. This was our behavioural outcome measure—their choice between a healthier prize (fruit) and an unhealthy one (chocolate). After choosing their reward, participants moved on to a debrief screen, which made it clear that the feedback given about life expectancies in their area had been false (debrief text is included in the questionnaire shown in Supplemental Information).

Covariates

The age and gender that the participants entered at the beginning of the experiment were used as covariates. Their postcode was used to generate a deprivation score for their current residential neighbourhood. This was done using the Office for National Statistics’ Indices of Multiple Deprivation (IMD; Mclennan et al., 2011). The IMD identify the most deprived areas of the country by combining a range of economic and social indicators into a single score. Areas can be identified by their IMD rank, which is considered to be a useful objective measure of an individual resident’s socioeconomic status (Danesh et al., 1999). We used the statistics for the lower layer super output areas—LSOAs. Finally, we used the lengths of time that the participant spent on the participant information screen and the priming screen as covariates. We did this because participants who spent more time reading the cover story and feedback information may have believed the cover story to a greater extent and thus may have been more strongly primed.

Analysis

All analysis was carried out in SPSS version 19. We excluded data from participants whose self-reported location was not consistent with our location checks (see Supplemental Information). The effects of our covariates on reported health intentions were assessed using a GLM. This was done so that any covariates that had a significant effect on self-reported health intentions could be controlled for in our main statistical model.

The effects of treatment on reward choice were evaluated using binary logistic regression. As in the GLM, we first assessed which, if any, of the covariates had an effect on reward choice in order to include them in the main model as needed. The data for all experiments reported in this paper can be accessed as part of the Supplemental Information.

Results

Descriptive statistics

35 participants were randomly allocated to the controllable long life treatment and 37 to the uncontrollable short life treatment. 39 participants were male and 33 were female. Ages ranged from 19 to 69 years. Time spent on the information page ranged from 0 to 199 s, with a mean of 20 s. Time spent on the priming pages ranged from 9 to 138 s, with a mean of 22 s. Participants’ neighbourhood IMD scores ranged from 3.64 to 65.40 (of a possible 0.53–87.80) with a mean of 23.88.

There was no significant difference in the ages of the participants across treatments (t70 = −0.50, p = 0.62). There was also no difference between treatments in the time spent on the information page (t69 = 0.70, p = 0.48) or the priming page (t69 = 1.09, p = 0.28). The IMD score of participants’ postcodes did not vary across treatments (t61 = −0.59, p = 0.558). There was no difference in the distribution of the sexes of participants across treatments (Fisher’s exact, p = 0.35).

Main results

There was no effect of any of our covariates on self-reported health intentions. Thus, the covariates were not included in the main model (Table 1). There was also no effect of treatment on the self-reported health intentions (Tables 1 and 2).

Table 1 GLM results for experiment 1.

GLM results showing the effect of the covariates (model 1) and the controllable long life and uncontrollable short life treatments (model 2) on self-reported health intentions.

	F	p	ηp2	
Model 1: covariates only a				
Age	1.44	0.238	0.115	
Sexb	0.72	0.585	0.061	
IMD score	0.37	0.828	0.033	
Time on info page	1.65	0.178	0.131	
Time on priming page	1.58	0.196	0.126	
Model 2: model for treatment effect b , c				
Treatment	1.47	0.223	0.093	
Notes.

a df = 4, error = 44, p = significance (∗p ≤ 0.05).

b The reference category is female.

c df = 4, error = 57, p = significance (∗p ≤ 0.05).

Table 2 Means for experiment 1.

Means and standard deviations for self-reported health intentions in the controllable long life and uncontrollable short life treatments.

Reported health intention	Treatment	Mean
(standard deviation)	
Effort in looking after health	Uncontrollable short life	62.67 (26.72)	
Controllable long life	67.93 (20.96)	
Intention to eat 5 portions
of fruit and veg per day	Uncontrollable short life	47.94 (34.29)	
Controllable long life	63.17 (26.80)	
Intention to exercise three
times over the coming week	Uncontrollable short life	60.70 (33.82)	
Controllable long life	56.03 (31.85)	
Intended units of alcohol
intake over the coming week	Uncontrollable short life	5.69 (7.08)	
Controllable long life	8.03 (16.18)	

None of the covariates showed an effect on choice of fruit, rather than chocolate, as a reward. However, there was an effect of treatment on reward choice (Table 3). Of the participants in the uncontrollable short life treatment, 31% (n = 10) chose fruit as a reward. In the controllable long life treatment, 57% (n = 20) of the participants chose fruit (Fig. 1, Table 3).

Table 3 Binary logistic regression results for experiment 1.

Binary logistic regression results showing the effect of the covariates (model 1) on the odds ratios for selecting fruit over chocolate and the effect of the controllable long life prime compared with the uncontrollable short life prime (model 2).

	Odds ratio
(lower CI–upper CI)	p	
Model 1: covariates only			
Sexa	1.64 (0.54–5.01)	0.383	
Age	1.01 (0.97–1.06)	0.653	
Neighbourhood deprivation score	1.00 (0.96–1.03)	0.896	
Time spent on information page	1.00 (0.97–1.04)	0.790	
Time spent on priming page	0.96 (0.91–1.01)	0.128	
Model 2: model for treatment effect			
Treatment	2.93 (1.08–8.00)	0.036*	
Notes.

CI = 95% confidence interval, p = significance (∗p ≤ 0.05).

a The reference category is female.

Figure 1 Fruit and chocolate choice in experiment 1.

The percentage of participants who chose fruit or chocolate rewards after exposure to either a controllable long life prime or uncontrollable short life prime.

Experiment 1 discussion

Contrary to our prediction, the results of experiment 1 demonstrated no effect of our primes on self-reported health intentions. However, there was an effect of our primes on a health-related decision—the choice of fruit versus chocolate. The effect of treatment on reward choice was notable. The proportion of participants who chose fruit went up from 31% in the uncontrollable short life prime to 57% in the controllable long life treatment (an 84% relative increase). The fact that there was an effect of the prime on the behavioural measure but not the self-report measures suggests that the priming may produce an implicit, automatic response, rather than an explicit, reasoned one. This is interesting, given that prior evidence suggests that a number of health-related decisions involve implicit, automatic processes (Gibbons, Houlihan & Gerrard, 2009; Sheeran, Gollwitzer & Bargh, 2013).

Several aspects of experiment 1 needed improving upon. The experiment had no control condition, so we could not say what the baseline preferences with no priming would be. Our design also did not separate the effects of priming mortality per se from those of controllability, since our two primes differed in both these dimensions. Finally, it is possible that the effect seen in experiment 1 was actually a normative one: in the uncontrollable short life condition, the health behaviour of others was not mentioned. Meanwhile, in the controllable condition, the health behaviour of others was described. Social norms are thought to influence health behaviour (Ball et al., 2010; Wood, Brown & Maltby, 2012), and it is possible our participants were automatically conforming to the norms described in the primes. It was important to rule out this potential confound. Thus, in experiment 2, we added a control treatment, and designed new primes which separated the effect of mortality salience from that of controllability. Since the norms contained in the two controllable treatments were opposing, this also addressed the potential of a confounding normative effect.

Experiment 2: Separating The Effects of Mortality Priming From Those Of Controllability Priming

Our second online experiment built upon our first. We added a control condition in which participants entered their demographic data and postcode, but received no feedback about life expectancy for people in their demographic. We also separated out the life expectancy component of the message (whether it suggested that people were living for more or less time than others) from the controllability of the causes of mortality. Thus, there were five conditions: uncontrollable short life, uncontrollable long life, controllable short life, controllable long life and a control condition. Our Uncontrollable Mortality Risk Hypothesis (see Introduction) predicts that the controllability of the primed mortality risk should be more important than whether or not mortality per se is made salient. Thus, we hypothesized that participants in the two controllable treatments would be more likely to choose fruit than participants in the uncontrollable treatments, regardless of whether the prime suggested that people were living longer or dying younger. In light of the result of experiment 1, we expected that we might see no effect of treatment on self-reported health intentions.

Methods and materials

As in experiment 1, participants were recruited using Crowdflower and followed a link to a Qualtrics-based experiment. The experiment was launched on August 14, 2013. The participant information, consent form and location check screens were the same as those used in experiment 1 (see Supplemental Information). Again, participants entered their demographic information, saw a “loading” animation, and then were randomly allocated to one of the treatments. While the primes in experiment 1 were personalised to age, gender and postcode, experiment 2 primes were only personalised by postcode. In addition, the reference frames were changed. We did this in order to test a form of words which would not involve deceit, because in our later field study (experiment 3, see below), there would be no opportunity to debrief participants. This meant shifting the reference frame (either the same residential area in the year 2000, or other UK regions in the present), so that deceit was not necessary (because it is true that people in Tyne & Wear are living longer than they were in the year 2000, but also, not as long as others in the UK—see experiment 3).

Control condition

In the control condition, there was no feedback after the participant entered their information. They simply waited for 12 s at the loading screen and then saw the message, “Thanks for submitting your basic information. Please answer the following questions about your health”.

Uncontrollable short life prime

The uncontrollable short life prime consisted of a message saying that people living in the participant’s postcode area were dying younger than people in other parts of England. The reasons given for this were beyond the participant’s control—in this case, high rates of violent crime and traffic accidents: “Statistics indicate that, on average, people in your postcode area [(postcode)] die younger than people in other parts of England. This seems to be because there are higher rates of traffic accidents and violent crime than in other areas. Please answer the following questions about your health”.

Uncontrollable long life prime

The uncontrollable long life prime said that people living in the participant’s postcode area, were now living longer than they had in the year 2000. Again, the reasons given were beyond individual control: “Statistics indicate that, on average, people in your postcode area [(postcode)] are living longer now than they were in the year 2000. This seems to be because of improvements in road safety and reductions in violent crime. Please answer the following questions about your health”.

Controllable short life prime

The controllable short life prime stated that people living in the participant’s postcode area, were dying younger than people in other parts of England. This time reasons given were within individual control—in this case, individual health behaviours: “Statistics indicate that, on average, people in your postcode area [(postcode)] die younger than people in other parts of England. The reasons for this are unclear, but it may be due to individual behaviours, such as diet and exercise habits. We want to understand more about why this is happening. Please answer the following questions about your health”.

Controllable long life prime

The controllable long life prime consisted of a message saying that people living in the participant’s postcode area, were now living longer than they had in the year 2000. Again, the reasons given were controllable: “Statistics indicate that, on average, people in your postcode area [(postcode)] are living longer now than they were in the year 2000. The reasons for this are unclear, but it may be due to individual behaviours, such as diet and exercise habits. We want to understand more about why this is happening. Please answer the following questions about your health”.

Outcome variables

The outcome variables were the same as those used in experiment 1.

Covariates

As in experiment 1, age, gender, postcode IMD score and time spent on the information and priming pages were used as covariates.

Exclusions

The exclusion criteria were the same as those used in experiment 1 (see Supplemental Information for details).

Analysis

As in experiment 1, the effects of our covariates on reported health intentions were assessed using a GLM, so that any that had a significant effect could be included in the main model. We also used custom contrasts to investigate whether there were differences between the uncontrollable and controllable treatments and between the long and short life treatments.

As in experiment 1, the effects of treatment on reward choice were tested using binary logistic regression. Again, we first assessed whether any covariates had an effect on reward choice, so that they could be included in our model. We ran a factorial treatment model, which contrasted the effects of the controllable treatments with the uncontrollable and the long life treatments with the short life ones.

Results

Descriptive statistics

There were 35 participants in the control treatment, 59 in the uncontrollable short life treatment, 44 in the uncontrollable long life treatment, 31 in the controllable short life treatment and 26 in the controllable long life treatment. There were 117 male participants and 78 female. Ages ranged from 18 to 73 years. Time spent on the information page ranged from 1 to 1,402 s, with a mean of 102 s. Time spent on the priming pages ranged from 0 to 448 s, with a mean of 19 s. IMD scores ranged from 3.15 to 87.80 (of a possible 0.53–87.80) with a mean of 25.84.

There was no significant difference in the ages of the participants across treatments (F4,190 = 1.20, p = 0.31). There was no difference between treatments in the time spent on the information page (F4,184 = 0.69, p = 0.60) or the priming page (F4,186 = 1.78, p = 0.13). There was also no significant difference in the IMD score of participants’ postcodes across the treatments (F4,170 = 0.99, p = 0.414). The distribution of the sexes of the participants was not significantly different across treatments (Fisher’s exact, p = 0.13).

Main results

In our covariates only model, there was an effect of sex on self-reported health intentions. Specifically, there was an effect of sex on intention to exercise (Table 4), with males having a greater intention to exercise than females (male mean = 70.34, s.e. = 2.97; female mean = 58.13, s.e. = 3.50). Thus, sex was included in the main model. However, as in experiment 1, there was no effect of treatment on self-reported health intentions (Tables 4 and 5). There were also no significant differences in reported health intentions when we compared controllable with uncontrollable or long life with short life conditions using custom contrasts (Table 6).

Table 4 GLM results for experiment 2.

GLM results for the effect of covariates on health intentions (model 1) and the adjusted model for treatment plus sex, which had a significant effect in the first model (model 2).

Model 1: covariates only a	F	p	ηp2			
Age	1.05	0.384	0.040			
Sex	3.30	0.014*	0.116			
IMD score	1.22	0.305	0.046			
Time on info page	0.35	0.844	0.014			
Time on priming page	0.50	0.735	0.019			
Model 2: Model for treatment effect b	F	p	ηp2	df	df error	
Treatment	1.01	0.437	0.032	12	363	
Sex	4.92	0.001*	0.142	4	119	
Notes.

a df = 4, error = 101, p = significance (∗p ≤ 0.05).

b p = significance (∗p ≤ 0.05).

Table 5 Means for experiment 2.

Means and standard deviations for self-reported health intentions in experiment 2.

Self-reported intentions	Treatment	Mean
(standard deviation)	
Effort in looking after health	Control	67.24 (24.14)	
Uncontrollable long life	67.63 (21.91)	
Uncontrollable short life	62.53 (21.57)	
Controllable long life	65.4 (28.40)	
Controllable short life	60.26 (26.29)	
Intention to eat 5 portions
of fruit and veg per day	Control	50.84 (31.13)	
Uncontrollable long life	60.94 (27.67)	
Uncontrollable short life	52.4 (29.20)	
Controllable long life	67.73 (25.88)	
Controllable short life	57.17 (31.96)	
Intention to exercise three
times over the coming week	Control	60.6 (33.99)	
Uncontrollable long life	69.13 (29.92)	
Uncontrollable short life	66.53 (30.76)	
Controllable long life	57.40 (38.94)	
Controllable short life	62.52 (31.41)	
Intended units of alcohol
intake over the coming week	Control	6.64 (9.84)	
Uncontrollable long life	6.88 (7.75)	
Uncontrollable short life	5.55 (9.82)	
Controllable long life	3.07 (3.90)	
Controllable short life	3.13 (5.83)	

Table 6 Custom contrast results for experiment 2.

Results of custom contrasts between controllable and uncontrollable, and short and long life treatments for self-reported health intentions.

	Sum of
squares	Mean
square	F	p	
Custom contrast of controllable versus uncontrollable
conditions					
Effort in looking after health	101.41	101.41	0.18	0.672	
Intention to eat 5 portions of fruit and veg per day	26.53	26.53	0.03	0.861	
Intention to exercise three times over the coming week	1022.65	1022.65	0.99	0.322	
Intended units of alcohol intake over the coming week	63.45	63.45	0.68	0.410	
Custom contrast of long life versus short life conditions					
Effort in looking after health	1266.21	1266.21	2.25	0.135	
Intention to eat 5 portions of fruit and veg per day	1528.08	1528.08	1.77	0.185	
Intention to exercise three times over the coming week	323.19	323.19	0.31	0.577	
Intended units of alcohol intake over the coming week	64.55	64.55	0.70	0.406	
Notes.

df = 1, p = significance (∗p ≤ 0.05).

None of the covariates in the covariates only model had an effect on choice of fruit as a reward (Table 7). Thus, no covariates were included in the main model. There was an effect of treatment on reward choice. Participants in the controllable treatments were more likely to choose fruit than participants in the uncontrollable treatments, or in the control (Table 7, Fig. 2). However, there was no difference in food choice between the short and long life primes (Table 7, Fig. 2). That is, there was an effect of the controllability of the mortality risk that was primed. The effect was of a similar magnitude to that seen in experiment 1. In the control treatment, 55% (n = 18) chose fruit. In the uncontrollable treatments 51% and 51% (uncontrollable long life, n = 21 and uncontrollable short life, n = 29) of participants chose fruit. In the controllable treatments, 71 and 75% (controllable long life, n = 15, controllable short life, n = 20) of the participants choose fruit.

Table 7 Binary logistic regression results for experiment 2.

Binary logistic regression results showing the effect of covariates and of treatments on the odds of selecting fruit over chocolate.

	Odds ratio
(lower CI–upper CI)	p	
Model 1: covariates only			
Sexa	0.68 (0.30–1.50)	0.340	
Age	1.03 (0.99–1.07)	0.125	
Neighbourhood deprivation score	1.00 (0.98–1.03)	0.978	
Time spent on information page	1.03 (0.99–1.06)	0.134	
Time spent on priming page	1.00 (0.99–1.01)	0.470	
Model 2: model for treatment effect			
Controllable vs. uncontrollable	2.59 (1.22–5.47)	0.013*	
Long life vs. short life	1.06 (0.54–2.10)	0.862	
Notes.

CI = 95% confidence interval, p = significance (∗p ≤ 0.05).

a The reference category is female.

Figure 2 Fruit and chocolate choice in experiment 2.

The percentage of participants who chose fruit or chocolate rewards in response to controllable or uncontrollable, long or short life primes and the control condition of experiment 2.

Experiment 2 discussion

Experiment 2 parsed the effects of controllability from those of long and short life primes. The results showed that people were more likely to choose fruit over chocolate in the controllable, but not the uncontrollable treatments, regardless of whether they were told they were likely to have longer, or shorter life spans. The result in the experimental control treatment looked similar to those in the uncontrollable treatments (Fig. 2). This suggests that, at least for the sample of participants in experiment 2, the “default” reward preference was akin to the preference under conditions of uncontrollable mortality.

As in experiment 1, there was no effect of treatment on self-reported intentions, but there was an effect on reward choice. As discussed for experiment 1, this suggests an implicit or automatic decision process, rather than an explicit or reasoned one.

The results of experiment 2 helped us to rule out the possibility that the effect seen in experiment 1 was a normative one. In experiment 1, in the uncontrollable short life condition, the health behaviour of others was not mentioned. Yet, in the controllable long life condition, it was the health behaviour of others in the participants’ demographic that was suggested to be the cause of their longevity. This might have elicited a social norms effect by suggesting that others of the same demographic were living healthy lives. Norms are thought to play a role in influencing health behaviour (Ball et al., 2010; Wood, Brown & Maltby, 2012). Thus, it was important that we use experiment 2 to rule out the possibility of a normative effect. In experiment 2, in the controllable mortality condition, the norm was that people were dying younger because of poor health habits. The selection of fruit still increased in this condition, relative to the uncontrollable and control conditions, suggesting that the result of experiment 1 was not due to a normative effect.

Although experiment 2 parsed the effects of controllability from those of long and short life primes and also ruled out the possibility of a normative effect, another potential confound remained: there may have been a demand effect, because both experiments 1 and 2 were explicitly health related. In order to rule this out, we ran a third experiment in the field.

Experiment 3: A Replication Of The Controllability Priming Effect In A Surreptitious Field Experiment

This field experiment built upon our online experiments. We ran it as a surreptitious experiment in order to remove any demand characteristics. This also allowed us to test whether the effect could be seen in a real-world setting. The study took place in a busy shopping centre in the Tyne and Wear area. Participants were told that they were taking part in a public opinion survey run by Newcastle University in exchange for being entered into a prize draw. Rather than our participants giving their details and receiving feedback about the average person of their demographic, we primed them using a question on the polling card. The questions suggested that people in Tyne and Wear are living longer, either due to uncontrollable causes, or due to controllable ones. That is, the primes were both long life primes, but the controllability of the causes was different. We hypothesised that, as in experiments 1 and 2, participants in the controllable treatment would choose fruit more often than participants in the uncontrollable treatment.

Methods

Recruitment

Participants were recruited at a large shopping centre in the Tyne and Wear area. Data were collected over two weekends in November 2013, with the first run of data collection running from Friday to Sunday and the second on a Saturday and Sunday (five days in total). The experimenter stood next to a pop-up stand with two large polling boxes and the prize draw cards. The pop-up stand and the cards gave instructions for participating. The experimenter also explained the entry procedure verbally. Participants were asked to complete a polling card with their name, address and date of birth. They were then asked to circle their answer to a multiple choice question (the prime—see details below) and to place their card into a polling box. The main incentive to participate was the chance of winning one of three £100 shopping vouchers. Participants were told that they would all be entered for the chance to win this main prize. As “bonus” prizes there were ten organic fruit boxes and ten chocolate collection boxes to be won. Participants had to indicate which of these they would prefer to win, by posting their card into the relevant polling box. The primes were presented alternately at the polling stand in two hour slots, which were counterbalanced across the 50 h during which data was collected.

Covariates

Age was calculated from the date of birth entered on the polling cards. As in the two online experiments, postcode IMD score was also used.

Primes

We used two primes, both longevity-focussed, but differing in their controllability. In the uncontrollable condition, participants were asked to answer the following multiple choice question: “Recent statistics show that people in Tyne and Wear are living longer now than they were in the year 2000. Why do you think this is? (A) Because there are fewer traffic accidents. (B) Because there is less violent crime. (C) Both: there are fewer traffic accidents and less violent crime”. This question was designed to imply that the most important local sources of mortality were things beyond individual control. In the controllable condition, participants were asked to answer a different multiple choice question: “Recent statistics show that people in Tyne and Wear are living longer now than they were in the year 2000. Why do you think this is? (A) Because people have more control over the kind of healthcare they receive. (B) Because people are looking after themselves better. (C) Both: people have more control over their care and are looking after themselves better”. This question was intended to imply that the most important local sources of mortality were things within individual control. (An electronic copy of the prize draw card can be found in Supplemental Information.)

Outcome variable

The outcome variable was our participants’ choice of bonus prize. As in experiments 1 and 2, this could be either an organic fruit box worth £11 or a chocolate collection box worth £11.

Analysis

As in experiments 1 and 2, the effects of treatment on reward choice were evaluated using binary logistic regression. In model 1 we assessed the effects of the covariates, so that any that had a significant effect could be included in the model for treatment effect (model 2).

Results

Descriptive statistics

There were 121 participants in the uncontrollable treatment, and 116 in the controllable treatment. Ages ranged from 15 to 87 years. IMD scores ranged from 3.75 to 74.48 (of a possible 0.53–87.80) with a mean of 27.91.

There was no significant difference in the ages of the participants across treatments (t229 = −0.78, p = 0.43). There was also no significant difference in the IMD score of participants’ postcodes across the treatments (t227 = −0.16, p = 0.875).

Main results

Neither age, nor neighbourhood IMD score had any effect in the covariates only model. Thus, they were not included in the main model (Table 8). There appeared to be an effect of treatment on tendency to choose fruit, as a reward. Of the participants in the uncontrollable treatment, 22% (n = 27) chose fruit as a reward. In the controllable treatment, 34% (n = 39) of participants chose fruit, a 54% relative increase (Fig. 3). However, the result of the binary logistic regression was marginally non-significant (p = 0.054, Table 8).

Table 8 Binary logistic regression results for experiment 3.

Adjusted model showing the odds of selecting fruit over chocolate by experimental treatment with the uncontrollable treatment as the reference category.

	Odds ratio
(lower CI–upper CI)	p	
Model 1—covariates only			
Age	1.01 (1.00–1.03)	0.177	
Neighbourhood deprivation score	1.00 (0.98–1.02)	0.825	
Model 2—model for treatment effect			
Treatment	1.76 (0.99–3.14)	0.054	
Notes.

CI = 95% confidence interval, p = significance (∗p ≤ 0.05).

Figure 3 Fruit and chocolate choice in experiment 3.

The percentage of participants who chose fruit or chocolate rewards in response to controllable or uncontrollable long life primes.

Experiment 3 discussion

Our field experiment replicated the pattern seen in our online experiments, although the effect was marginally non-significant. This may have been due to a lack of power to detect the effect, which was smaller than in the other studies (odds ratios: experiment 1 = 2.93; experiment 2 = 2.59; experiment 3 = 1.76). However, given that qualitatively similar results were found for all three studies, we can be more confident that the statistically marginal result of experiment 3 represents a real effect (Moonesinghe, Khoury & Janssens, 2007). Future experiments should use larger samples to ensure adequate power.

There were some ways in which the effects seen in experiments 1 and 2 may have been diluted in experiment 3. The uncontrolled nature of the experimental environment allowed unpredicted participant behaviours. For example, some participants (n = 13) filled out the question card and then handed the card a child or spouse, allowing them to choose the prize (invariably the children chose chocolate). Once the cards were in the polling boxes, they could not be traced, so these participants could not be identified or excluded from the analysis. If participants had not allowed those who accompanied them to choose the prizes, the effect might have been larger, but unfortunately it is not possible to confirm this.

Similarly, the fact that the experiment took place in a large shopping centre during November may have influenced the results. Many participants were at the centre to do their Christmas shopping. When selecting chocolate, some participants (number not noted) made comments such as, “I would choose fruit for myself, but chocolate will make a good Christmas present for someone”. Thus, the effect might have been diluted in this experiment, but not in the online experiments, which were carried out earlier in the year.

There was one other minor issue with the field experiment (3). The experimenter was not blind to the treatments. However, the online experiments (1 and 2) were double-blind, since the treatments were randomly allocated by Qualtrics, and, as we have seen, the results were comparable.

The fact that the observed effect was replicable in a surreptitious experiment goes some way towards ruling out the possibility of a demand effect. Participants were not aware that they were taking part in an experiment, or that it was related to health behaviour.

Finally, the result of experiment 3 demonstrates that the effect seen in the online experiments can be translated into a real world setting. This suggests that enhancing people’s sense of control over sources of mortality and ill health could be an effective way of improving real world health behaviours.

Overall Discussion

The results of our online and field experiments lend support to the Uncontrollable Mortality Risk Hypothesis. They suggest that perceptions about the controllability of mortality risk may have an important influence on health behaviours. Experiment 1 was the first, to our knowledge, to demonstrate an effect of uncontrollable mortality priming on a health-related decision. Experiment 2 was the first to separate out the effects of uncontrollable and controllable mortality primes on a health-related decision. Experiment 3 replicated the main effect of the first two experiments in a surreptitious experiment, suggesting that the effect seen in the first two experiments was not due to any demand characteristic.

While our experimental treatments affected participant behaviour, there was no effect on our participants’ self-reported intentions (experiments 1 and 2). This implies that the decision to take fruit as a reward may have involved implicit and automatic processes (occurring without explicit reasoning see Evans, 2003), even when health was made salient. That is, people may not consciously calculate their degree of control over their mortality risk and then decide whether to choose a healthy or unhealthy reward. Previous research shows that a number of health behaviours seem to involve implicit processes and there have been calls to examine the role of implicit processes in health behaviour more closely (Gibbons, Houlihan & Gerrard, 2009; Sheeran, Gollwitzer & Bargh, 2013).

In our introduction, we outlined theoretical perspectives that shared features of the Uncontrollable Mortality Risk Hypothesis. Although our experiments were not designed to test the predictions of the alternative hypotheses outlined in our introduction, we can still discuss our results in their context.

Our results may help to shed light on the associations between Health Locus of Control and health behaviour (Reitzel et al., 2013; Wardle & Steptoe, 2003). When people feel that they have low control in general (external control beliefs), they are likely to believe that they have little control over their mortality risk. If so, investing effort, time or money in controlling what little they can, would have a lower payoff than for others who feel that they have more control over their mortality risk (internal control beliefs).

The Extended Parallel Process Model states that messages depicting threats will be acted upon to the extent that the available solutions are seen to be effective (Witte & Allen, 2000). It proposes that a threat must have severe consequences in order to gain people’s attention and motivate them to act. In addition to this, the recommended action must be perceived to be highly effective for this motivation to be translated into behavioural change. However, our result suggests that a threat does not need to be overt for an effect to be seen. In our experiments, there were no dramatic fear appeals. We simply mentioned that people of the participant’s demographic were either living longer (or not) than average and manipulated the causes to be more or less controllable. In experiment 3, health was barely mentioned and no health advice was given. Nonetheless, we saw a switch to a healthier reward choice. This is likely to be because the choice was between two foods which are widely known to be healthy (fruit) and unhealthy (chocolate). No further health information was needed. This demonstrates that fear appeals may not be necessary to motivate behaviour change. In some cases, where the healthy choice is widely known to be so (e.g., to not smoke), recommended health actions may not be needed. It may be enough simply to reduce perceived (or better still, actual) uncontrollable mortality risks. Indeed, the fact that uncontrollable mortality risk alters the likely payoff of investing in health, could help to explain why interventions intended to improve health behaviours simply by giving information have been ineffective (e.g., Buck & Frosini, 2012; Downs et al., 2013). Merely giving information could be insufficient to change motivation (Pepper & Nettle, 2014b; White, Adams & Heywood, 2009), especially when the information given only pertains to risks already perceived as controllable and does nothing to reduce the severity of any uncontrollable risks perceived.

If the effects of our primes were implicit and automatic, as they appeared to be, this would contradict the predictions of the Terror Management Health Model. The Terror Management Health Model predicts that people should act in a health oriented way when explicitly primed, but not when the mortality salience is implicit (Goldenberg & Arndt, 2008). In addition, in the treatments where participants were told they would live longer than average, it could be reasoned that mortality is made more distant, rather than salient. However, we still saw an effect in these treatments, based on whether the causes of mortality were controllable, rather than upon whether premature mortality was emphasised.

More research on the effects of uncontrollable mortality risk is needed. If mortality controllability priming could be used to increase motivation towards healthy behaviours, then it is important to test it in new populations and situations and to learn more about when it works. For example, our primes were effective in a situation where people were being offered a food reward free-of-charge. However, the situation may be different when people are paying for the food themselves. Our reward options were binary (fruit versus chocolate). Results may be different if there is a range of options to choose from—especially if the options are less obviously healthy and unhealthy ones. Furthermore, the experiments we have run so far have only examined food choice. We do not currently know whether such primes can be used to influence other health-related decisions. Finally, although this is beyond the scope of the hypothesis, it is possible that control over factors other than mortality risk may influence health behaviour. Future experiments could include additional treatments, which prime the controllability of risks unrelated to mortality, such as the risks of becoming unemployed or becoming a victim of theft.

It is also important to learn more about perceptions of the controllability of common mortality risks. Understanding where perceptions come from could help policy makers to influence any sources of information which lead to misconceptions. For example, if media scare stories bias perceptions of uncontrollable mortality risk, then increasing awareness of this issue among journalists and calling for increased journalistic responsibility would be important.

The effect of controllability may go beyond health behaviour. It is possible that the controllability of mortality risk influences a range of behaviours involving trade-offs between costs and rewards in the present and those in the future. When the risk of death is high (and cannot be mitigated), the odds of being alive to receive future rewards are reduced. Thus, people who believe they have a high and uncontrollable risk of mortality should be less future-oriented than those who believe that they can control their mortality risk. There is some support for this idea in the existing literature. Differences in time perspective have been shown to be associated with a variety of health behaviours (Adams & Nettle, 2009; Adams & White, 2009; Adams, 2009), and with differences in reproductive scheduling (Daly & Wilson, 2005; Kruger, Reischl & Zimmerman, 2008; Pepper & Nettle, 2013). There is also evidence to suggest that differences in time perspective could be caused by exposure to signals of mortality risk. For example, future discounting has been found to be steeper in people who had experienced a larger number of recent bereavements (Pepper & Nettle, 2013) and in recent earthquake survivors, compared to controls (Li et al., 2012).

The results of our experiments support the idea that perceptions about the controllability of mortality risk may be an important factor influencing people’s health-related decisions. This finding is congruent with other evidence about the importance of Health Locus of Control for health (Burker et al., 2005; Holt et al., 2000; Poortinga, Dunstan & Fone, 2008; Wardle & Steptoe, 2003; Williams-Piehota et al., 2004) and the influence of mortality priming on behaviour (Griskevicius et al., 2011a; Griskevicius et al., 2011b; Mathews & Sear, 2008). However, our Uncontrollable Mortality Risk Hypothesis is subtly different to other perspectives in the health literature and the results of our experiments suggest that the difference may be a crucial one.

Adjusting perceptions about the controllability of mortality risk could become an important tool in health interventions. Our findings also emphasise the importance of tackling sources of mortality which are beyond individual control. Making neighbourhoods and work places safer would have the primary benefit of reducing mortality risks beyond individual control, but could also lead to improved health behaviours.

Supplemental Information

Supplemental Information 1 Supplement

This supplement contains additional detail on the methods used in the experiments reported in the Pepper & Nettle paper, “Out of control mortality matters: the effect of perceived uncontrollable mortality risk on a health-related decision.”

The supplementary information includes: (1) Details on the methods used to ensure the data used were from participants who had been honest about their location. (2) The participant information, consent, questionnaire and debrief used in experiments 1 and 2, and (3) the prize draw cards used in experiment 3.

Click here for additional data file.

Supplemental Information 2 Data from experiment 1

Click here for additional data file.

Supplemental Information 3 Data from experiment 2

Click here for additional data file.

Supplemental Information 4 Data from experiment 3

Click here for additional data file.

We acknowledge Benjamin Wilson, Jean Adams and Stephanie Clutterbuck for their informal peer review of the design of the experiments. We would also like to thank Intu Properties plc, for allowing us to collect data at one of their shopping centres, and our participants, for taking part in the experiments.

Additional Information and Declarations

Competing Interests

Author Contributions

Human Ethics

The authors declare there are no competing interests.

Gillian V. Pepper conceived and designed the experiments, performed the experiments, analyzed the data, contributed reagents/materials/analysis tools, wrote the paper, prepared figures and/or tables, reviewed drafts of the paper.

Daniel Nettle analyzed the data, contributed reagents/materials/analysis tools, reviewed drafts of the paper.

The following information was supplied relating to ethical approvals (i.e., approving body and any reference numbers):

All of our experiments received ethical approval from the Newcastle University Faculty of Medical Sciences ethics committee (reference: 00554).

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
