# Peer review of "Out of control mortality matters: the effect of perceived uncontrollable mortality risk on a health-related decision"

_PeerJ, doi:10.7717/peerj.459_

## Round 0.1 · original submission · Minor Revisions

Dear authors. I read your ms with great interest and agree with the reviewer that your manuscript would be suitable for publication given some minor amendments. In particular I strongly encourage you to follow the suggestion to condense and shorten your paper. Moreover the suggested recommendation for future research could be made more explicit (eg larger samples).

·

Basic reporting

I think there is too much detail on the various theoretical frameworks presented in the introduction with seemingly similar predictions, and I would suggest substantially reducing this section. I would also suggest a re-writing the section of the Uncontrollable Mortality Risk Hypothesis (UMRH); the summary paragraph 148-154 includes most of the relevant information and the rest should instead be referred to the theoretical model (Nettle, 2010). On the other hand, I would welcome a little more on evolutionary perspective of the main hypothesis, specifically on the proximate and ultimate causes of the evolution of the behaviour (as described on lines 113-116). Do you argue it's maladaptive? Or do instead believe it’s part of an adaptive strategy to increase RS in an unstable environment?

The methods section is again perhaps too detailed. The article would improve if a more concise version was provided in the main text, with the rest added to the SI. However, do keep the priming questions in the main text.

If you ran the full models with the co-variates, are the treatments still significant? If that’s the case do you think this is just because of lack of power? Ideally you should present the full model with all theoretically valid variables included (at least in the SI)

Minor Points:
- Remove bracket before Kong on line 70
- Please briefly allude to what Kong & Shen, 2011 found in their experiment.
- Please mention you used Super Output Areas (SOAs) as your neighbourhood geographic unit, if that's the case.
- I think the legends of Table 1 and 2 are inversed.
- Tables should be more concise, put SD and 95% CI in brackets with the mean and odds ratio, respectively, and present the units of the means.
- Missing dates for when experiment 2 was conducted.
- Show graphs in colour to improve legibility.
- Missing “to” in between “used” and “increase” in line 721
- There should not be a significant asterisk for Treatment on Table 4 as it’s only marginally significant

Experimental design

The controllable primes you used are arguably explicit primes, as in experiments 1 and 2 they mention diet, so you may be directly priming the participants to want to eat healthier food. In experiment 3, there’s a slightly less explicit priming, but “looking after themselves” is still easily connected to making a healthier choice. As a result, this does not specifically test the UMRH (as distinguished from other hypotheses, described in lines 132-140), as the same prediction would apply to other hypotheses. This should be added as a limitation in the discussion, and “unconnected health-related decision” in line 665 and “primes seemed to be implicit and automatic” in line 712 appear to me as an over-statement.

I'm aware that there are two schools of thought in relation to lying to experiment participants, but this is problematic and should be avoided, as increases distrust of participants in future experiments (i.e. pollutes the sample). In an Economics journal this would be enough reason for this study to not be published. Nevertheless, I understand that providing the true life expectancies would have been logistically more complex and you would have required a larger sample, but nevertheless you seem to have done that in experiment 2 and 3.

Not a crucial point, but why use IMD rank as a co-variate instead of IMD scores? Neighbourhoods with similar scores can have very different ranks, so I think it would be more accurate to use scores.

Validity of the findings

The UMRH and your tests are specifically related to mortality cues, but wouldn’t it be likely that any type of exposure to risk produce similar results? Why would humans have a specialised reaction to mortality, instead of a more general heuristic? In order to test that specifically mortality cues are priming behaviour and not other ‘bad’ things happening, you would have to compare the mortality cue with other negative cues, for example controllable and uncontrollable criminality or morbidity cues. I would suggest mentioning this in the discussion.

Additional comments

Overall, I think the studies presented are very interesting and in particular the 3rd experiment, which would be great if it could be replicated with a larger sample size.

---

## Round 0.2 · accepted · Accept

Dear authors. Thank you for making the effort to shorten and improve your ms. I am happy to accept it in its current form.